# The Prevalence and Determinants of Hesitancy for Regular COVID-19 Vaccination among Primary Healthcare Patients with Asthma or COPD in Greece: A Cross-Sectional Study

**DOI:** 10.3390/vaccines12040414

**Published:** 2024-04-14

**Authors:** Izolde Bouloukaki, Antonios Christodoulakis, Stavroula Papageorgakopoulou, Ioanna Tsiligianni

**Affiliations:** 1Department of Social Medicine, School of Medicine, University of Crete, 71003 Heraklion, Greece; achristodoul@hmu.gr (A.C.); medp2012079@med.uoc.gr (S.P.); i.tsiligianni@uoc.gr (I.T.); 2Department of Nursing, School of Health Sciences, Hellenic Mediterranean University, 71410 Heraklion, Greece

**Keywords:** COVID-19, vaccination, COPD, primary healthcare

## Abstract

The emergence of novel coronavirus mutations and signs of the waning immunity provided by COVID-19 vaccines underscore the importance of regular vaccination. This study aimed to investigate the prevalence of regular COVID-19 vaccination hesitancy and the factors that influence it among patients with asthma or chronic obstructive pulmonary disease (COPD) who visited primary care centers. The cross-sectional study was conducted in six primary healthcare centers in Crete, Greece (October–December 2023). Participants completed a questionnaire, which included questions about socio-demographic characteristics, health status, previous COVID-19 booster vaccination, attitudes, and beliefs about COVID-19 vaccination. Multivariate logistic regression was used to identify the factors that influenced vaccine hesitancy. Of the 264 participants, 65% exhibited hesitancy towards COVID-19 booster vaccination. Female gender, middle age, lower educational attainment, depression diagnosis, concerns about vaccine side effects, lack of confidence in vaccine efficacy, and reliance on media information were positively associated to vaccine hesitancy. Conversely, those having a cardiovascular disease or type 2 diabetes, those generally adhering to the recommendations of the treating physician, and those having previously received the flu vaccine or more than three COVID-19 vaccine doses were positively associated with regular COVID-19 vaccinations. Consequently, our findings could help develop strategies that could potentially reduce regular vaccination hesitancy among patients with asthma or COPD.

## 1. Introduction

The emergence of COVID-19 in December 2019 has posed a significant global health threat, profoundly affecting people’s lives, especially those living with chronic diseases [1,2]. Chronic lung diseases, such as asthma and chronic obstructive pulmonary disease (COPD), have been identified as risk factors for unfavorable outcomes associated with severe COVID-19 infection, including hospitalization and mortality [3,4,5,6,7,8,9,10]. Vaccination against COVID-19 has been developed as the principal strategy to prevent the rapid spread of COVID-19 and mitigate the morbidity associated with the disease [11,12]. However, patients with chronic respiratory diseases may experience less-effective immune responses to vaccination, as their humoral responses are less effective than those of healthy individuals [13,14], leading to an increased risk of infection and hospitalization [15]. Nevertheless, patients with COPD seem to experience an increase in both airway and systemic immune responses following the administration of a third dose [16], while individuals with asthma or COPD who receive booster doses (third and fourth doses) also benefit from this additional protection against the COVID-19 infection, resulting in reduced risks of COVID-related mortality, hospitalizations, and severe complications [17]. The aforementioned benefits highlight the importance of implementing regular vaccination strategies for this specific group of patients. In response, the GOLD recommendations for stable COPD and the GINA recommendations for asthma have been revised to incorporate COVID-19 vaccination as a preventive measure starting in 2022 [18,19].

A significant challenge in achieving widespread COVID-19 vaccination is vaccine hesitancy [20,21], with rates reaching up to 30% [22]. Vaccination hesitancy is defined as the “delay in acceptance or refusal of vaccination despite availability of vaccination services” [23]. Factors such as reluctancy and uncertainty towards receiving the first dose of the COVID-19 vaccine can be indicative of the acceptability of booster doses [24]. Moreover, studies have indicated that the COVID-19 primary vaccination course uptake rate among patients with COPD (39.0–86.57%) and asthma (73.6–88.0%) was variable [25,26,27,28]. It is surprising that a considerable proportion of unvaccinated patients were not advised by their doctors to get vaccinated against COVID-19, or they had inadequate knowledge about the vaccine and were uncertain about its effectiveness and safety [25]. Nevertheless, there is limited information regarding patients with COPD or asthma who hesitate to receive regular COVID-19 vaccinations. One recent study showed that the coverage of COVID-19 booster vaccination among patients with COPD was less optimal (62.45%) [29] than the target of 70% suggested by the World Health Organization (WHO) [30]. To the best of our knowledge, there have been no reports on the coverage of COVID-19 booster vaccinations among patients with asthma.

The European Center for Disease Prevention and Control, as well as studies conducted in Greece, reports that the majority of adults have received a primary vaccination course (at least two doses) of the COVID-19 vaccine [31,32,33,34]. This is probably attributed to the strict measures imposed on unvaccinated individuals in Greece, excluding them from indoor activities such as dining in restaurants or bars, participating in-person at scientific conferences, attending cinemas, entering stadiums, or visiting museums, resulting in their social exclusion, with those over the age of 60 facing the additional burden of monthly fines for half a year. Another study from Crete, Greece, on vaccine hesitancy towards the COVID-19 vaccine in Greek primary care settings already indicated the presence of fear associated with the primary vaccination course [33]. More research in Greece regarding regular COVID-19 vaccination showed that a variable percentage of people (ranging from 47.5% to 74.8%) [34,35] had already received regular COVID-19 vaccinations, while 60% expressed their willingness [36] to receive booster doses.

At present, there is limited knowledge about the prevalence and factors that affect regular COVID-19 vaccination among patients with COPD or asthma in Greece. Understanding people’s willingness to receive regular COVID-19 vaccines is an urgent requirement, especially for patients with chronic respiratory diseases, given the dynamic nature of the pandemic, the likelihood of viral mutations, and the potential decline in immunity. The healthcare community, especially primary care providers, has a crucial responsibility in promoting and influencing vaccine acceptance within the communities they serve. Therefore, this study aimed to examine the prevalence and factors influencing hesitation towards receiving regular COVID-19 vaccinations among patients with asthma or COPD within primary care settings in Greece.

## 2. Materials and Methods

### 2.1. Study Design and Population

A cross-sectional study was conducted in six primary healthcare centers in Crete, Greece, over a three-month period from October to December 2023. All patients with a confirmed diagnosis of asthma or COPD (*n* = 287) who were treated in these primary healthcare centers during usual consultations were approached, informed, and subsequently asked for their voluntary participation. The inclusion criteria included being over 18 years old, having a physician diagnosis of asthma or COPD, and providing written informed consent. Conversely, the exclusion criteria were patients with severe neurological or mental diseases or those with poor understanding of the Greek language. Considering the aforementioned criteria, a total of 264 patients (92% response rate) were finally included in the study. The investigators approached the participants after the regular consultation and administered a questionnaire that included socio-demographic characteristics, health status factors, previous COVID-19 vaccination, and attitudes and beliefs about COVID-19 vaccination.

### 2.2. Data Collection

Based on a previous publication [33] and the relevant literature [25,26,27,28,29], a structured questionnaire was developed (Appendix A). The questionnaire consisted of questions from different categories: (1) socio-demographic factors, such as age, gender, marital status, and educational background; (2) factors influencing health status, including smoking habits, previous medical conditions, self-assessed health, vulnerability level, and the vulnerability of family members; (3) information on their COVID-19 vaccination status, including the number of doses received and any side effects reported; (4) willingness to receive additional (booster) doses of the vaccine (yes/no); (5) attitudes and beliefs regarding the COVID-19 vaccine (20 items with questions of agreement or disagreement); (6) level of knowledge and questions about the impact of various factors on vaccine participation, such as religion, politics, science, media, and the anti-vaccine movement and whether they had received the influenza vaccination during the previous season; and (7) concerns regarding the possibility of getting infected with COVID-19. To evaluate the health status of patients with asthma or COPD, the Asthma Control Test (ACT) and COPD Assessment Test (CAT), respectively, were employed. The ACT questionnaire is a self-administered instrument that healthcare professionals use to identify patients with inadequately controlled asthma. The ACT questionnaire consists of five Likert-scale items, rated from one to five, that assess different aspects of asthma: (1) limitations in daily activities, (2) breathlessness, (3) waking up due to asthma symptoms, (4) use of rescue medication, and (5) overall assessment of asthma. The total ACT scores were calculated by adding up the scores from the five items, which could range from 5 (indicating poor asthma control) to 25 (indicating complete asthma control). Higher scores are indicative of improved asthma control. An ACT score of 19 or lower suggests uncontrolled asthma [37,38]. The COPD Assessment Test (CAT) is a questionnaire designed to assess the self-reported effects of COPD on an individual’s health status [39]. The CAT questionnaire includes eight elements (cough, phlegm production, chest tightness, breathlessness, limited activities, confidence in going outside, sleeplessness, and energy) that the patient assesses on a scale from 0 to 5. The score ranges between 0 and 40, with higher values indicating worse health status. A threshold of 10 or higher was used to indicate poor health status.

### 2.3. Statistical Analysis

The results are presented in the form of the mean ± standard deviation (SD) for continuous variables with a normal distribution, and as the median (25th–75th percentile) for variables without a normal distribution. Qualitative variables are represented as absolute numbers or percentages. To compare the hesitant and non-hesitant patient groups regarding vaccination, we utilized a two-tailed *t*-test for independent samples (assuming normal distribution) or a Mann–Whitney U test (for abnormally distributed data) for continuous variables. For categorical variables, we employed Pearson’s chi-square test. To determine the factors influencing hesitancy towards regular COVID-19 vaccination, a multivariate logistic regression analysis was conducted. The analysis was adjusted for confounding variables, such as age, gender, smoking status, education level, and co-morbidities. We performed calculations to obtain the odds ratios (ORs) and their accompanying 95% confidence intervals (95% CIs). To assess the presence of multicollinearity, we utilized collinearity statistics to examine the correlations between predictor variables. The tolerance value and variance inflation factor indicated that the collinearity was within an acceptable range. Age was examined as both a continuous variable and as categorical groups, divided into 18–49, 50–64, and ≥65 years. In the context of this analysis, the term “cardiovascular disease” (CVD) was employed as a predictor in logistic regression models, referring to a range of conditions, including coronary disease, atrial fibrillation, TIA/stroke, and heart failure. The results were deemed significant if the *p*-values were less than 0.05. We used SPSS software (version 25, SPSS Inc., Chicago, IL, USA) to analyze the data.

## 3. Results

### 3.1. Study Population

The study population included 264 participants (92% response rate), of whom 54% (*n* = 142) were men. The ages of participants ranged from 19 to 88 years; 7% (*n* = 18) were aged 18–49, 35% (*n* = 92) were aged 50–64, and 58% (*n* = 154) were aged ≥65 years. About 36% of patients had a primary school-level education or less. The socio-demographic characteristics and health status factors of the 264 participants are further described in Table 1. Interestingly, more than half of the participants (68%, *n* = 180) rated their health as good/excellent.

Regarding chronic respiratory diseases, 62% of the patients (*n* = 164) had COPD, and 38% (*n* = 100) had asthma. Patients diagnosed with COPD, when compared to those with asthma, tended to be older and were mostly males, with higher rates of primary education, married status, current smoking, and higher prevalence of hypertension, cardiovascular disease, and type 2 diabetes. In contrast, individuals diagnosed with asthma exhibited a higher prevalence of inflammatory arthritis. The majority of participants diagnosed with COPD were categorized into either group A (38%) or B (38%), while group E accounted for 24% based on the GOLD 2023 classification [37]. In the 12-month period prior to the study, the majority (82%) of patients did not suffer from COPD exacerbations or had just one, whereas 18% experienced two or more exacerbations, and 6% required hospitalization. Participants with asthma were classified as “well controlled” (66%) or “not well controlled” (34%) based on their ACT scores. In the 12-month period prior to the study, the majority (78%) of patients did not suffer from asthma exacerbations or had just one, whereas 22% experienced two or more exacerbations, and 4% of those that experienced exacerbations required hospitalization.

Among the 264 participants, a subgroup of individuals (8%, *n* = 20) had never received a COVID-19 vaccine. However, most of the participants (66%) had received at least three doses of the vaccine, with 9% (*n* = 24), 5% (*n* = 14), and 4% (*n* = 10) of individuals receiving fourth, fifth, and sixth doses, respectively. All patients were provided with the same vaccine, namely the Pfizer-BioNTech COVID-19 vaccine. Out of the 264 participants, 14% (*n* = 38) reported experiencing side effects, which encompassed a wide range of symptoms, such as mild allergic reactions, flu-like symptoms, shortness of breath, fatigue, nausea, dizziness, muscle weakness, and muscle pain.

### 3.2. Hesitancy towards Regular COVID-19 Vaccination

In total, 65% of the individuals participating in the study expressed hesitancy towards receiving the COVID-19 vaccine on a yearly basis. Among those with asthma, this percentage was 76%, while among patients with COPD, it was 59% (*p* < 0.001). The prevalence of hesitancy exhibited a reverse V-shaped trend among the study participants across age groups (Figure 1). Hesitancy was the lowest among younger participants aged 18–49 years (44%), followed by a significant increase among middle-aged participants aged 50–64 years (83%). Subsequently, hesitancy decreased among participants aged over 65 years, reaching a rate of 57%. Interestingly, the prevalence of hesitancy varied across different age groups among asthma patients, with a notable increase as patients aged, reaching a plateau after the age of 65 (Figure 1).

Other baseline characteristics of the study population regarding vaccination hesitancy are shown in Table 2. The two groups did not differ significantly in terms of gender, age, level of education, marital status, smoking status, or self-rated health status. However, there were differences in the way the co-morbidities presented. The non-hesitant group showed a significantly higher presence of CVD (9% vs. 22%, *p* = 0.003) and type 2 diabetes (33% vs. 20%, *p* = 0.02), whereas the hesitant group exhibited a higher prevalence of depression (16% vs. 7%, *p* = 0.02).

### 3.3. Determinants of Hesitancy for Regular COVID-19 Vaccination

Table 3 and Table 4 present a comparison of the experiences, attitudes, and beliefs about the COVID-19 virus and the vaccination rates of individuals who are hesitant vs. non-hesitant towards COVID-19 booster vaccination. Higher vaccination doses and flu vaccination history were linked to less hesitancy towards COVID-19 vaccination. Vaccine hesitancy tended to be more prevalent among individuals who expressed concerns about the potential side effects, observed side effects in their social circle, doubted the effectiveness of the vaccine, and believed that natural infection offered better immunity than vaccination.

Table 5 and Table 6 depict the proportions of individuals within the asthma or COPD populations classified by clinical status and their respective preferences for vaccination. A higher willingness to be vaccinated was observed among patients with asthma who had a better clinical status (higher ACT score). No significant variation was found in the clinical status of COPD patients between those who exhibited hesitation and those who did not (*p* > 0.05).

After controlling for confounding variables, the multivariate logistic regression analysis (Table 7) showed that females were almost twice as likely as males to report hesitancy [OR: 1.91 (1.03–3.56), *p* = 0.04]. Additionally, age continued to have a negative correlation with hesitancy [OR: 0.97 (0.94–0.99), *p* = 0.023], with adults between 50 and 64 years old having higher rates of hesitancy compared to other age groups [OR: 4.627 (1.493–14.335), *p* = 0.008]. Participants having lower educational status were also more likely to be hesitant [OR: 1.848 (1.004–3.402), *p* = 0.04]. CVD [0.320 (0.159–0.642), *p* = 0.001] and type 2 diabetes [OR: 0.530 (0.287–0.979), *p* = 0.04] remained as negative correlates of reporting hesitancy, whereas the presence of depression [OR: 3.196 (1.223–8.352), *p* = 0.018] was positively associated with COVID-19 vaccine hesitancy. Previous COVID-19 vaccination [OR: 0.149 (0.073–0.305), *p* < 0.001] and previous flu vaccination [OR: 0.403 (0.195–0.834), *p* = 0.014] also had a negative correlation with reporting hesitancy. Participants who expressed concerns about vaccine side effects [OR: 85.688 (31.420–233.689), *p* < 0.001] and had low confidence in the vaccine’s effectiveness [OR: 8.127 (1.830–36.090), *p* = 0.006] were more inclined to exhibit hesitancy. A recommendation from treating physicians [OR: 0.041 (0.018–0.095), *p* < 0.001] was linked to less hesitancy, whereas media/internet information was associated with higher odds of hesitancy [OR: 20.247 (7.016–58.434), *p* < 0.001]. There was no association found between clinical status and hesitancy among patients with COPD, as indicated by the CAT scores [OR: 1.012 (0.956–1.071, *p* = 0.682] and GOLD classifications of B [OR: 0.865 (0.420–1.782, *p* = 0.695] or E [OR: 0.723 (0.316–1.652), *p* = 0.441]. However, among patients with asthma, worse clinical status based on the ACT [OR: 0.799 (0.651–0.979), *p* = 0.031] was associated with vaccination hesitancy.

## 4. Discussion

This cross-sectional study assessed the prevalence of hesitancy towards receiving regular COVID-19 vaccinations among patients with asthma or COPD who visited primary care facilities in Crete, Greece, as well as the factors that influenced this hesitancy. We found that a significant proportion of these patients expressed hesitancy towards receiving booster doses of the COVID-19 vaccine. Our study also identified several factors that contributed to hesitancy towards regular COVID-19 vaccinations, including female gender, middle age, lower educational attainment, depression diagnosis, concerns about vaccine side effects, lack of confidence in vaccine efficacy, and reliance on media information. Conversely, having a cardiovascular disease or type 2 diabetes, generally adhering to the recommendations of the treating physician, and having previously received the flu vaccine or having more than three COVID-19 vaccine doses were positively associated with regular COVID-19 vaccinations.

In our study, we examined patients with asthma or COPD, the two most frequently encountered chronic respiratory conditions in primary care settings [38]. Despite COPD and asthma being distinct chronic respiratory diseases with differing causes, prognoses, and treatment approaches, they frequently exhibit common triggering factors, including respiratory infections [17,39]. Ensuring regular COVID-19 vaccination among these individuals is of utmost importance, as there is a clear association between COVID-19 infections and heightened morbidity and mortality rates [7,8,9]. Moreover, among these patients, there are indications of declining immunity provided by the initial COVID-19 vaccine series and also the emergence of new COVID-19 variants [13,14]. Interestingly, our study found that asthma patients who had worse clinical status were more likely to be hesitant about getting vaccinated. This aligns with a recent study that found a lower COVID-19 vaccination rate among individuals with poorly controlled asthma and those using biologic therapies [27]. The patients with COPD and asthma in primary care settings exhibited a hesitancy rate of 65% towards receiving regular COVID-19 vaccinations, which is higher than previous research on the general population in Greece (38.1%) [36] and around the world [24,40,41,42,43,44,45,46,47,48,49] and the global average COVID-19 booster vaccination hesitancy rate of 30.72% [22]. Our findings indicate that individuals with asthma or COPD are less willing to get a second booster compared to their willingness to receive the first booster dose. More specifically, 66% of our population had already been given three doses of the vaccine, compared to the 9% who had a fourth dose. Furthermore, these results are in line with a recent study which demonstrated that 72% of patients suffering from COPD or asthma received their initial booster dose, while only 9.6% received a second booster dose [17].

It is widely acknowledged that vaccination intentions and behaviors are subject to change over time, as evidenced by studies conducted at various stages of the pandemic that clearly demonstrated fluctuations in attitudes towards the COVID-19 vaccines [50]. In our study, various socio-demographic traits were found to be associated with vaccine hesitancy. As shown in other studies, being female was found to increase the likelihood of vaccine hesitancy [36,43,51,52,53,54,55,56,57,58]. Older age, especially in the range of 50–64 years, also played a significant negative role in determining patient acceptance of the COVID-19 vaccine, which is similar to other studies [40,41,52]. Furthermore, it seems that apart from older-aged individuals, vulnerable populations, such as patients at higher risk of clinical complications, have exhibited a decreased willingness to receive vaccinations due to heightened concerns about potential side effects [59]. We also identified that having lower level of education was associated with lower vaccine acceptance, which is also supported by findings from a previous systematic review that associated higher acceptance rates of the COVID-19 vaccine with higher levels of education [43,52]. Moreover, there is evidence suggesting that individuals with a higher level of education have exhibited a heightened perception of the risks associated with COVID-19 infection [60]. This could be attributed to their access to multiple reliable sources of information, which in turn affects their overall knowledge and awareness of COVID-19 vaccines [61].

Concerns about vaccine side effects, lack of confidence in the vaccine’s effectiveness, and reliance on media/internet information were all factors contributing to COVID-19 vaccine hesitancy. It is not surprising that doubts about vaccine effectiveness have grown due to the introduction of new vaccines and the rapid advancements in vaccine technologies [62,63]. Our findings are consistent with prior research indicating that the majority of participants believed booster doses to be ineffective and declined vaccination due to concerns about the vaccine’s side effects [40,43,52,64,65]. The presence of adverse events after primary vaccination emerged as the primary indicator for hesitancy in receiving the booster dose. The concern surrounding this issue may be attributed to the belief that the booster dose has the potential to cause more severe adverse reactions than the standard vaccine doses [65]. Additionally, in a previous study, it was found that participants’ primary reasons for being hesitant about receiving the COVID-19 vaccine booster dose were their skepticism regarding the scientific evidence supporting the benefits of the booster dose and the requirement for a booster dose at least once a year, given that it was administered recently [42]. These observations may be associated with earlier case reports [66,67,68,69] which showed a rise in adverse respiratory events among populations with respiratory diseases after COVID-19 vaccination. Nevertheless, emerging evidence suggests that COVID-19 vaccines can be given to patients with COPD or asthma without any safety concerns and with good efficacy [17]. Interestingly, in this study, social media played a crucial role as the primary source of information on COVID-19 and vaccination, which supports previous research [33]. This suggests that the access to unverifiable media information had an effect on patients’ choices and behavior [70], thereby potentially negatively influencing vaccination acceptance [71], especially among patients with chronic respiratory diseases, such as in our population. In Greece, the promotion of COVID-19 vaccination by authorities and medical associations has declined compared to the earlier days of the pandemic. This can add to the problem of people being hesitant to get vaccinated, underscoring the urgent need for these relevant authorities to strengthen their efforts in this regard.

Our study also identified that having CVD or type 2 diabetes, receiving more than three COVID-19 vaccine doses, having previously received the flu vaccine, and adhering to the recommendations of the treating physician were positively associated with lower vaccine hesitancy. Similarly, previous studies have demonstrated a correlation between prior COVID-19 and influenza vaccinations and a higher likelihood of booster acceptance [41,52]. Our results are also consistent with other studies showing that having chronic respiratory diseases, and patient trust in scientists and medical professionals regarding the vaccine’s effectiveness, significantly increased willingness to accept vaccines and booster doses [43]. On the other hand, the likelihood of vaccine hesitancy was found to be higher among our patients who were diagnosed with depression, which is consistent with earlier studies showing that people with depressive symptoms were more hesitant to receive COVID-19 booster doses [72]. In addition, our study emphasizes the crucial role of trustworthy informational sources, such as recommendations from the treating physician, as opposed to relying solely on media or internet sources.

The findings of this study emphasize the importance of achieving higher vaccination coverage for COVID-19 booster doses, as the current low rates could result in severe outcomes such as increased COVID-19 infections, a decrease in available healthcare staff, and a surge in COVID-19 hospitalizations. Considering the crucial role of healthcare professionals in primary care, it is reasonable to assume that actively involving primary care providers could enhance vaccine acceptance in communities [73]. In addition, considering the potential for individuals to change their minds from their initial intentions to their ultimate decisions [74], it becomes necessary to implement a personalized approach to ensure that individuals are making informed choices. Effective training programs are necessary to enable healthcare professionals to effectively communicate in a clear and understandable way with patients regarding COVID-19 vaccination and provide patients with comprehensive information. A recent study conducted in Greece focused on developing a pilot eLearning intervention to address COVID-19 vaccine hesitancy among primary healthcare and social services professionals [75]. This training had a positive impact on providers, as 90% of them expressed their intention to incorporate the information and skills acquired into their clinical practice.

It should be noted that, to the best of our knowledge, this is the first study that has investigated hesitancy towards regular COVID-19 vaccination and the factors that could influence it among patients with asthma or COPD in Greece. Nevertheless, certain limitations of the present study must be addressed. First, acceptance and hesitancy towards vaccines change over time as new evidence and vaccination data emerge. Consequently, the results of this study may also change over time. Second, causal inferences cannot be made due to the study’s cross-sectional design. In order to better understand individuals’ attitudes towards COVID-19 booster vaccinations and how they may change over time, it is important to conduct mixed-method longitudinal studies. Third, the subjects were recruited through a non-random sampling technique, which may have implications for the generalizability and external validity of the findings. Fourth, self-reported data may be subject to recall bias and a predisposition to present socially desirable responses. Fifth, it is important to note that our study was conducted in Crete, one of the 13 regions of Greece. When compared to the other Greek regions, the region of Crete surpasses the national average in health, life satisfaction, and employment, although it falls slightly below the national average in education [76]. Therefore, we cannot generalize our findings to other Greek populations.

## 5. Conclusions

In conclusion, a substantial proportion of individuals with COPD or asthma expressed hesitancy towards receiving regular COVID-19 vaccinations. Several factors were identified that could contribute to this hesitancy, including female gender, middle age, fear of vaccine side effects, co-morbidities, the history of previous vaccinations, lack of confidence in the vaccine’s efficacy, and the influence of scientists and media/internet. Therefore, our findings could provide useful insights to healthcare managers and healthcare professionals regarding COVID-19 booster vaccinations. These insights could help design more targeted campaigns and immunization programs/interventions to promote COVID-19 booster vaccinations and reduce vaccination hesitancy in these populations. Consequently, by increasing COVID-19 booster vaccinations, immunity to COVID-19 infection could be improved, thus reducing the morbidity and mortality of patients with asthma or COPD.

## Figures and Tables

**Figure 1 vaccines-12-00414-f001:**
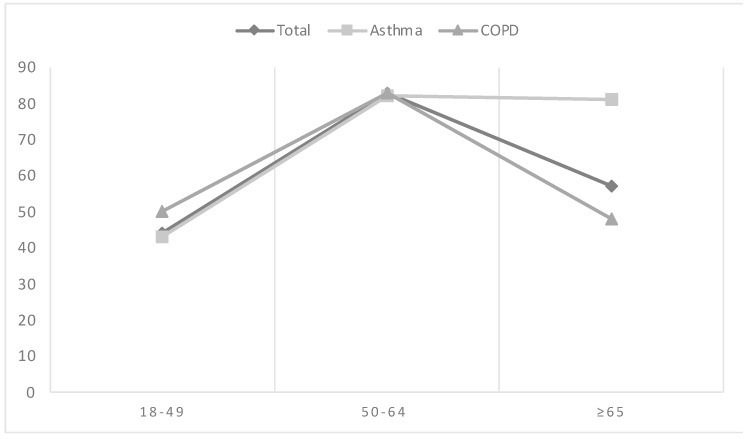
The estimated prevalence of hesitancy for regular COVID-19 vaccination by respiratory condition.

**Table 1 vaccines-12-00414-t001:** Characteristics of all 264 participants categorized based on respiratory disease.

	Total Population (*n* = 264)	Asthma (*n* = 100)	COPD (*n* = 164)	*p*-Value
**Demographics**				
**Gender, male (%)**	142 (54%)	34 (34%)	108 (66%)	<0.001
**Age, years**	65 ± 12	60 ± 13	69 ± 9	<0.001
**Age > 60 years**	180 (68%)	50 (50%)	130 (79%)	<0.001
**Level of education**				
Primary level or less	96 (36%)	28 (28%)	68 (42%)	0.023
Secondary level or higher	176 (64%)	72 (72%)	94 (58%)	
**Marital Status**				
Married	210 (80%)	70 (70%)	140 (86%)	0.001
Single	52 (20%)	30 (30%)	22 (14%)	
**Smoking status**				
Current	96 (36%)	18 (18%)	78 (48%)	<0.001
Never/former smoker	168 (64%)	82 (82%)	86 (52%)	
**Co-morbidities**				
Hypertension	155 (59%)	49 (49%)	106 (65%)	0.012
Cardiovascular disease	74 (28%)	18 (18%)	56 (34%)	0.005
Type 2 diabetes	64 (24%)	12 (12%)	52 (32%)	<0.001
Cancer	18 (7%)	2 (3%)	16 (11%)	0.04
Inflammatory arthritis	26 (10%)	20 (20%)	6 (4%)	<0.001
Gastroesophageal reflux disease	52 (20%)	24 (24%)	28 (17%)	0.17
Depression (on medications)	34 (13%)	14 (14%)	20 (12%)	0.67
Anxiety disorder	16 (6%)	8 (8%)	8 (5%)	0.30
**Self-rated health**				
Good	180 (68%)	68 (68%)	112 (68%)	0.27
Fair	80 (30%)	32 (32%)	48 (29%)	
Bad	4 (2%)	0 (0%)	4 (3%)	

**Table 2 vaccines-12-00414-t002:** Characteristics of the 264 participants regarding hesitancy towards COVID-19 booster vaccination.

	Total Population (*n* = 264)	COVID-19 Non-Hesitant (*n* = 92)	COVID-19 Hesitant (*n* = 172)	*p*-Value
**Demographics**				
**Gender, male (%)**	142 (54%)	56 (61%)	86 (50%)	0.09
**Age, years**	65 ± 12	67 ± 11	65 ± 12	0.09
**Age > 60 years**	180 (68%)	70 (76%)	110 (64%)	0.04
**Level of education**				
Primary level or less	96 (36%)	30 (33%)	66 (39%)	0.48
Secondary level or higher	176 (64%)	62 (67%)	106 (61%)	
**Marital Status**				
Married	210 (80%)	74 (80%)	138 (80%)	0.93
Single	52 (20%)	18 (20%)	34 (20%)	
**Smoking status**				
Current	96 (36%)	28 (30%)	68 (40%)	0.29
Never/Former smoker	168 (64%)	64 (70%)	104 (60%)	
**Co-morbidities**				
Hypertension	155 (59%)	53 (58%)	102 (59%)	0.79
Cardiovascular disease	74 (28%)	36 (39%)	38 (22%)	0.003
Type 2 diabetes	64 (24%)	30 (33%)	34 (20%)	0.02
Cancer	18 (7%)	10 (13%)	8 (6%)	0.09
Inflammatory arthritis	26 (10%)	10 (11%)	16 (9%)	0.706
Gastroesophageal reflux disease	52 (20%)	16 (17%)	36 (21%)	0.49
Depression (on medications)	34 (13%)	6 (7%)	28 (16%)	0.02
Anxiety disorder	16 (6%)	6 (7%)	10 (6%)	0.82
**Self-rated health**				
Good	180 (68%)	62 (67%)	118 (69%)	0.30
Fair	80 (30%)	30 (33%)	50 (29%)	
Bad	4 (2%)	0 (0%)	4 (2%)	

**Table 3 vaccines-12-00414-t003:** Reported experiences with COVID-19 virus and vaccination for individuals who were hesitant and non-hesitant towards COVID-19 booster vaccination.

Experiences	Total Population (*n* = 264)	COVID-19 Non-Hesitant (*n* = 92)	COVID-19 Hesitant (*n* = 172)	*p*-Value
**Previous COVID-19 infection**				
Yes	154 (58%)	52 (57%)	102 (59%)	0.66
No	110 (42%)	40 (43%)	70 (41%)	
**Severity of COVID-19 infection** **(in those previously infected, *n* = 154)**				
Mild	122 (80%)	40 (77%)	82 (80%)	0.93
Moderate	16 (10%)	6 (11.5%)	10 (10%)	
Severe	16 (10%)	6 (11.5%)	10 (10%)	
**COVID-19 vaccination doses**				
Zero	20 (7%)	4 (4%)	16 (9%)	<0.001
One	4 (2%)	2 (2%)	2 (1%)	
Two	18 (7%)	4 (4%)	14 (8%)	
Three	174 (66%)	48 (52%)	126 (73%)	
Four	24 (9%)	16 (18%)	8 (5%)	
Five	14 (5%)	8 (9%)	6 (4%)	
Six	10 (4%)	10 (11%)	0 (0%)	
**COVID-19 vaccination** **side effects (*n* = 38)**				
Mild	22 (58%)	6 (60%)	16 (57%)	0.63
Moderate/severe	16 (42%)	4 (40%)	12 (43%)	
**Flu vaccination**				
Yes	194 (74%)	78 (85%)	116 (67%)	0.002
No	70 (26%)	14 (15%)	56 (33%)	

**Table 4 vaccines-12-00414-t004:** Reported attitudes and beliefs about COVID-19 in groups that were hesitant and non-hesitant towards the COVID-19 booster vaccination.

	Total Population (*n* = 264)	COVID-19 Non-Hesitant (*n* = 92)	COVID-19 Hesitant (*n* = 172)	*p*-Value
**Fear of vaccine side effects**	148 (56%)	6 (7%)	142 (83%)	<0.001
**Reported vaccine side effects among family/friends**	32 (12%)	0 (0%)	32 (19%)	<0.001
**Low perceived efficacy of vaccine**	26 (10%)	2 (2%)	24 (14%)	0.002
**Early vaccine distribution**	2 (1%)	0 (0%)	2 (1%)	0.299
**Belief that other vaccines are also protecting for COVID-19**	6 (2%)	0 (0%)	6 (4%)	0.07
**No need due to previous COVID-19 infection/Belief that infection confers much greater immunity than a vaccine**	12 (5%)	0 (0%)	12 (7%)	0.018
**Perception of low susceptibility to disease or possible infection would not be severe**	6 (2%)	0 (0%)	6 (4%)	0.07
**Against vaccinations in general**	10 (4%)	4 (4%)	6 (4%)	0.727
**Fear of COVID-19 infection**	56 (21%)	22 (24%)	34 (20%)	0.43

**Table 5 vaccines-12-00414-t005:** Comparison of health status of patients with asthma between groups that were non-hesitant and hesitant towards COVID-19 booster vaccination.

	Total Population (*n* = 100)	COVID-19 Non-Hesitant (*n* = 22)	COVID-19 Hesitant (*n* = 78)	*p*-Value
**Patients with asthma**				
**ACT score**	20 ± 4	22 ± 3	20 ± 4	0.04
Well controlled (>19)	66 (66%)	16 (73%)	49 (63%)	0.39
Not well controlled (≤19)	34 (34%)	6 (27%)	29 (37%)	
**Exacerbations in the past year, *n* (%)**				
≤1	78 (78%)	13 (60%)	62 (69%)	0.44
≥2	22 (22%)	9 (40%)	16 (31%)	
≥1 hospitalization	4 (4%)	2 (9%)	2 (3%)	0.18

**Table 6 vaccines-12-00414-t006:** Comparison of health status of patients with COPD between groups that were non-hesitant and hesitant towards COVID-19 booster vaccination.

	Total Population (*n* = 164)	COVID-19 Non-Hesitant (*n* = 68)	COVID-19 Hesitant (*n* = 96)	*p*-Value
**Patients with COPD**				
CAT score	11 ± 6	11 ± 6	11 ± 6	0.98
CAT score ≥10	89 (54%)	36 (53%)	53 (55%)	0.76
**Exacerbations in the past year, *n* (%)**				
≤1	2 (82%)	52 (76%)	82 (85%)	0.08
≥2	6 (18%)	16 (24%)	14 (15%)	
≥1 hospitalization	10 (6%)	2 (3%)	8 (8%)	0.16
GOLD groups (*n*%)	6 (2%)	0 (0%)	6 (4%)	0.07
A	62 (38%)	24 (35%)	38 (40%)	0.79
B	62 (38%)	26 (38%)	36 (37%)	
E	40 (24%)	18 (27%)	22 (23%)	

**Table 7 vaccines-12-00414-t007:** Multivariate logistic regression analysis of factors associated with COVID-19 vaccine hesitancy.

Variables	Adjusted OR (95% CI)	*p*-Value
**Socio-demographic factors**		
Females vs. males	1.915 (1.030–3.561)	0.04
Age, years	0.97 (0.94–0.99)	0.02
Age group18–49	1	
Age group 50–64	4.627 (1.493–14.335)	0.008
Age group ≥ 65 years	1.151 (0.385–3.442)	0.80
Primary education vs. secondary/higher	1.848 (1.004–3.402)	0.04
Single vs. married	1.120 (0.520–2.413)	0.77
Current/former vs. no smoking	1.371 (0.657–2.857)	0.40
**Health status factors**		
Hypertension	1.324 (0.750–2.337)	0.33
Cardiovascular disease	0.320 (0.159–0.642)	0.001
Type 2 diabetes	0.530 (0.287–0.979)	0.04
Cancer	0.492 (0.179–1.353)	0.169
Inflammatory arthritis	0.630 (0.246–1.613)	0.336
Depression (on medications)	3.196 (1.223–8.352)	0.018
Self-rated health fair/bad vs. good	0.863 (0.484–1.536)	0.616
Self-rated as vulnerable group	1.229 (0.570–2.653)	0.599
**Experiences**		
Previous COVID-19 infection	1.076 (0.631–1.837)	0.788
Moderate/severe COVID-19 infection	0.543 (0.235–1.257)	0.154
COVID-19 vaccination Doses (>3 doses)	0.149 (0.073–0.305)	<0.001
COVID-19 vaccination side effects	1.571 (0.709–3.482)	0.265
Flu vaccination	0.403 (0.195–0.834)	0.014
**Attitudes**		
Fear of vaccine side effects	85.688 (31.420–233.689)	<0.001
Reported vaccine side effects among family/friends	Ν/A	
Low perceived efficacy of vaccine	8.127 (1.830–36.090)	0.006
Belief that infection confers much greater immunity than a vaccine	N/A	
Against vaccinations in general	0.481 (0.116–1.986)	0.311
Fear of COVID-19 infection	0.759 (0.400–1.442)	0.40
**Sources of information influencing COVID-19 uptake**		
Living with vulnerable groups	1.268 (0.640–2.513)	0.495
Doctors’ recommendation/science opinion	0.041 (0.018–0.095)	<0.001
Media/internet	20.247 (7.016–58.434)	<0.001

## Data Availability

The data that support the findings of this study are available from the corresponding author upon reasonable request.

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
