# Peer review of "The Prevalence and Determinants of Hesitancy for Regular COVID-19 Vaccination among Primary Healthcare Patients with Asthma or COPD in Greece: A Cross-Sectional Study"

_vaccines, 2024, doi:10.3390/vaccines12040414_

Round 1

Reviewer 1 Report

Comments and Suggestions for Authors

Despite being of some interest, comparing to the relevance of the content the text of the article is too long and redundant and really needs to be significantly shortened for a better readability. Too many references (85!) and tables, which should be reduced in number and redesigned or, if maintained, included as supplementary materials. 

The authors should also state if all patients received the same vaccine, or different vaccines were administered.

Line 115: relevant literature, please detail.

Line 177: GOLD 2023 classification, please give reference.

The questionnaire might be included as supplementary material.

The article cannot be published in the present form.

Comments on the Quality of English Language

The text needs to be revised for ortographic faults and the poor quality of the English language. 

Author Response

Despite being of some interest, comparing to the relevance of the content the text of the article is too long and redundant and really needs to be significantly shortened for a better readability. Too many references (85!) and tables, which should be reduced in number and redesigned or, if maintained, included as supplementary materials.

Response: We are grateful for your comments and the opportunity to revise and resubmit the manuscript. We tried to shorten the manuscript and reduce the references number as suggested. However, the journal does not impose any constraints on the number of references. In order to address other reviewers’ request for more references, we made an effort to maintain a balanced approach to all comments. Furthermore, the significance of the information presented on the reluctance towards regular COVID-19 vaccinations and the potential factors affecting it among patients with asthma or COPD, specifically in Greece, justifies the extensive length of the manuscript and the incorporation of these tables.

The authors should also state if all patients received the same vaccine, or different vaccines were administered.

Response: Thank you for your comment. All the patients received the same vaccine, the Pfizer-BioNTech COVID-19 Vaccine. This was clarified in the revised manuscript.

Page 5, Line 187-188, “All patients were provided with the same vaccine, namely the Pfizer-BioNTech COVID-19 Vaccine.”

Line 115: relevant literature, please detail.

Response: Thank you for your comment. Relevant literature was added, as suggested.

Line 177: GOLD 2023 classification, please give reference.

Response: Thank you for your comment. Reference was added.

The questionnaire might be included as supplementary material.

Response: Thank you for your comment. The questionnaire was included as supplementary material, as suggested.

The article cannot be published in the present form.

Response: Thank you for your comment. The article was revised as suggested.

The text needs to be revised for ortographic faults and the poor quality of the English language.

Response: The manuscript was subjected to language editing by Dr. Candida Dawn Suffridge, an English native speaker, as per your suggestion.

Reviewer 2 Report

Comments and Suggestions for Authors

This is a really interesting and well conducted study from Crete. They appropriately acknowledge the extent to which findings might not be generalisable but a significant strength was the high response rate from the clinics that were studied. 
Comparator data for "Greece" was provided. I do not know to what extent people who live in Crete are similar to the Greek population overall or whether they have important differences.  For example, the level of education in this sample would be interesting to compare with Crete as a whole and with Greece as a whole. 

The only significant omission is that there is no information on the extent of promotion of booster Covid vaccine in Crete. My impression in New Zealand where I am writing from is that the boosters are available and they probably are recommended in some guidelines but that there has not been much active promotion. The paper discusses the negative sources of information that the participants may have been exposed to but there is no information on whether there was for example any government promotion of getting a booster. 

Two small details that I think need addressing below.

Line 71-76 needs to be clear that these were the restrictions placed on the patients in Crete.

Line 343 "for" should be "from"

Author Response

This is a really interesting and well conducted study from Crete. They appropriately acknowledge the extent to which findings might not be generalisable but a significant strength was the high response rate from the clinics that were studied.

Response: We are grateful for your comments and the opportunity to revise and resubmit the manuscript.

Comparator data for "Greece" was provided. I do not know to what extent people who live in Crete are similar to the Greek population overall or whether they have important differences.  For example, the level of education in this sample would be interesting to compare with Crete as a whole and with Greece as a whole.

Response: We thank the reviewer for the comment, according to the  to the OECD Territorial Reviews “Regional Policy for Greece Post-2020” (accessed 29 March 2024) the region of Crete surpasses the national average in health, life satisfaction, and employment, although it slightly falls below the national average in education (Crete average 70.5% vs Greek average 76.7%)]. Therefore, we have added this as a limitation in the revised manuscript.

Page 17, Lines 417-422: “Fifth, it is important to note that our study was conducted in Crete, one of the 13 regions of Greece. When compared to the other Greek regions, the region of Crete sur-passes the national average in health, life satisfaction, and employment, although it falls slightly below the national average in education [86]. Therefore, we cannot generalize our findings to other Greek populations.”

The only significant omission is that there is no information on the extent of promotion of booster Covid vaccine in Crete. My impression in New Zealand where I am writing from is that the boosters are available and they probably are recommended in some guidelines but that there has not been much active promotion. The paper discusses the negative sources of information that the participants may have been exposed to but there is no information on whether there was for example any government promotion of getting a booster.

Response: We thank the reviewer for these comments, unfortunately, we confirm that the situation in Greece is relatively the same as in New Zealand, meaning that it is recommended in the official guidelines, however, there is limited active promotion, especially, when compared to the first dose of the vaccine. We have added a relevant comment in the discussion section as a suggestion.

 Page 16, Line 371-375: “In Greece, the promotion of COVID-19 vaccination by authorities and medical associa-tions has declined compared to the earlier days of the pandemic. This can add to the problem of people being hesitant to get vaccinated, underscoring the urgent need for these relevant authorities to strengthen their efforts in this regard.”

Two small details that I think need addressing below.

Line 71-76 needs to be clear that these were the restrictions placed on the patients in Crete.

Response: Thank you for your comment. These restrictions were imposed on patients not just in Crete but throughout Greece. This was clarified in the revised manuscript.

Page 3, Line 71-76 “This is probably attributed to the strict measures imposed on unvaccinated individuals in Greece, excluding them from indoors activities such as dining in restaurants or bars, participating in-person at scientific conferences, attending cinemas, entering stadiums, or visiting museums, resulting in their social exclusion, with those over the age of 60 facing the additional burden of monthly fines for half a year.”

Line 343 "for" should be "from"

Response: Thank you for your comment. It was corrected.

Page 16, Line 337-340 “Furthermore, it seems that apart from older aged individuals, vulnerable populations, such as patients at higher risk of clinical complications, exhibited a decreased willing-ness to receive vaccinations due to heightened concerns about potential side effects [68].”

Reviewer 3 Report

Comments and Suggestions for Authors

This paper reports a cross-sectional data analysis study of the prevalence of regular COVID-19 vaccination hesitancy and the factors that influence it among patients with asthma or chronic obstructive pulmonary disease (COPD) who visited six primary care centers in Crete, Greece for the October-December 2023 period. Overall, the paper is well written and the study follows standard statistical protocols for such studies. Here are a couple of items to attend to in a revision.

First, on p. 6, the paragraph:

Regarding reported sources of information influencing COVID-19 annual uptake, the non-hesitant group mostly relied on the recommendations of their treating physician, whereas the hesitant group did so to a much lesser extent (94% vs 42%, p<0.001). On the other hand, media/internet information (47% vs 4%, p<0.001) emerged as a key driving force behind hesitancy for a significant number of individuals.

seems to be out of place, as these items are not contained in Table 2. Rather, they appear in Table 7 on p. 11.

Second, the contents of Section 4. Discussion are good. It would be good, however, to highlight somewhat more the comparisons of findings from this study with those in other studies. This would give more prominence to the unique findings and contributions of the present study. 

Author Response

This paper reports a cross-sectional data analysis study of the prevalence of regular COVID-19 vaccination hesitancy and the factors that influence it among patients with asthma or chronic obstructive pulmonary disease (COPD) who visited six primary care centers in Crete, Greece for the October-December 2023 period. Overall, the paper is well written and the study follows standard statistical protocols for such studies. Here are a couple of items to attend to in a revision.

Response: We are grateful for your comments and the opportunity to revise and resubmit the manuscript.

First, on p. 6, the paragraph:

Regarding reported sources of information influencing COVID-19 annual uptake, the non-hesitant group mostly relied on the recommendations of their treating physician, whereas the hesitant group did so to a much lesser extent (94% vs 42%, p<0.001). On the other hand, media/internet information (47% vs 4%, p<0.001) emerged as a key driving force behind hesitancy for a significant number of individuals.

seems to be out of place, as these items are not contained in Table 2. Rather, they appear in Table 7 on p. 11.

Response: Thank you for your comment. This information was omitted for clarity.

Second, the contents of Section 4. Discussion are good. It would be good, however, to highlight somewhat more the comparisons of findings from this study with those in other studies. This would give more prominence to the unique findings and contributions of the present study.

Response: Thank you for your comments. Our belief is that the majority of our findings were compared to the findings of earlier studies. However, we also tried to identify additional studies and this section is now enriched with more specific information about the comparisons of findings from this study with those in other studies.

Round 2

Reviewer 1 Report

Comments and Suggestions for Authors

The author responded adequately to the comments and did the proposed changes. The English language has been revised.

The article may now be published in the present form

Reviewer 3 Report

Comments and Suggestions for Authors

The revisions to this manuscript have responded to the previous review. I have no further suggestions for revision.